# FlexiReID: Adaptive Mixture of Expert for Multi-Modal Person Re-Identification

**Zhen Sun** [* 1]  **Lei Tan** [* 2]  **Yunhang Shen** [3]  **Chengmao Cai** [1]  **Xing Sun** [3]  **Pingyang Dai** [1]  **Liujuan Cao** [1]  **Rongrong Ji** [1 4]

## Abstract

Multimodal person re-identification (Re-ID) aims to match pedestrian images across different modalities. However, most existing methods focus on limited cross-modal settings and fail to support arbitrary query-retrieval combinations, hindering practical deployment. We propose FlexiReID, a flexible framework that supports seven retrieval modes across four modalities: rgb, infrared, sketches, and text. FlexiReID introduces an adaptive mixture-of-experts (MoE) mechanism to dynamically integrate diverse modality features and a cross-modal query fusion module to enhance multimodal feature extraction. To facilitate comprehensive evaluation, we construct CIRS-PEDES, a unified dataset extending four popular Re-ID datasets to include all four modalities. Extensive experiments demonstrate that FlexiReID achieves state-of-the-art performance and offers strong generalization in complex scenarios.

## 1. Introduction

Pedestrian re-identification (ReID) is a critical technology in computer vision, focused on matching individuals across different camera viewpoints. This capability is essential for various surveillance and security operations, leading to diverse applications in fields such as security, urban management, retail, and law enforcement. ReID tasks are classified into single-modal ReID and cross-modal ReID. Single-modal ReID focuses on retrieval between RGB images, depending on the extraction and matching of visual features. However, it encounters several challenges in practical applications,

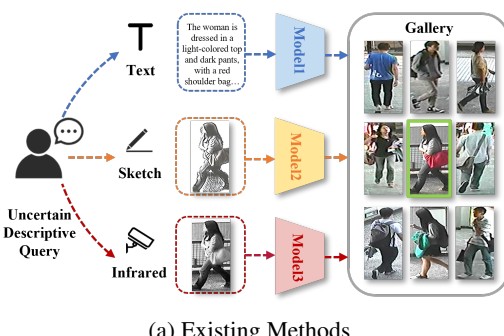

(a) Existing Methods

(b) Our Method

*Figure 1.* Illustration of our idea. Existing cross-modal ReID models primarily address single-modality retrieval, which limits their practical applicability. In contrast, our method enables flexible retrieval across seven different modality combinations. The green boxes match the query.

including variations in lighting, occlusions, differences in viewpoints, and changes in pedestrian poses, all of which can compromise recognition accuracy and robustness. In contrast, cross-modal pedestrian re-identification (ReID) presents significant advantages. Unlike single-modal approaches, cross-modal ReID incorporates various modalities, including textual descriptions, infrared images, and sketches. By leveraging multimodal information, cross-modal ReID offers complementary features that sustain high recognition accuracy across diverse environmental conditions, such as lighting variations and nighttime scenarios, thereby improving the system's robustness and adaptability. For example, in low-light or nighttime conditions, infrared images can provide clearer outlines of pedestrians compared

*Equal contribution [1]Key Laboratory of Multimedia Trusted Perception and Efficient Computing, Ministry of Education of China, Xiamen University, 361005, P.R. China. [2]National University of Singapore. [3]Tencent YouTu Lab. [4]Institute of Artificial Intelligence,Xiamen University,Xiamen,China. Correspondence to: Pingyang Dai <pydai@xmu.edu.cn>.

to RGB images, while textual descriptions and sketches can supply additional semantic and structural information when visual data is incomplete or ambiguous. However, existing cross-modal ReID, as shown in figure 1a , has an inherent limitation: it typically supports retrieval only between specific pairs of modalities, which restricts its ability to accommodate various combinations of modalities. In real-world scenarios, we often receive information from multiple modalities simultaneously. If retrieval is confined to a single modality paired with RGB images, the potential of existing multimodal information remains underutilized. This raises a critical question: **Can we establish a unified cross-modal person re-identification framework that supports flexible retrieval across any combination of modalities?**

Based on this, we propose the FlexiReID framework, which spans four modalities (Text, Sketch, Infrared, RGB) and supports seven different combinations of multimodal retrieval: Text-to-RGB, Sketch-to-RGB, Infrared-to-RGB, Text+Sketch-to-RGB, Text+Infrared-to-RGB, Sketch+Infrared-to-RGB, and Text+Sketch+Infrared-to-RGB, as shown in figure 1b. Inspired by CLIP, which is pre-trained on a large-scale dataset of 400 million image-text pairs, FlexiReID utilizes a simple dual-encoder architecture for the extraction of visual and textual features. Notably, the three visual modalities share a single image encoder. In order to efficiently extract features of diverse modalities, we introduce an Adaptive Expert Allocation Mixture of Experts (AEA-MoE) mechanism. Specifically, our proposed adaptive routing mechanism dynamically selects a varying number of expert combinations based on the attributes of the input features. Compared to the traditional Top-K routing mechanism, which selects a fixed number of experts, our approach better leverages the strengths of the multi-expert system, thereby optimizing the extraction of multi-modal features. Additionally, we designed a multi-modal feature fusion module called Cross-Modal Query Fusion (CMQF). This module accepts multi-modal feature inputs and uses learnable embedded features to compensate for missing modalities. Its superior feature fusion capability further enhances the flexible retrieval performance of FlexiReID. In this study, We introduce the concept of flexible retrieval to the field of person re-identification for the first time, pioneering a new research direction. The core idea of flexible retrieval is to accurately perform person retrieval using the existing modalities, even in the presence of missing modalities. To achieve this challenging objective, We expanded the modalities of the of the four datasets (CUHK-PEDES, ICFG-PEDES, RSTPReid, and SYSU-MM01), constructing a unified dataset named CIRS-PEDES, which encompasses four modalities: text, sketches, RGB images, and infrared images. Experimental results on the CIRS-PEDES show that the proposed FlexiReID outperforms several other state-of-the-art methods, demonstrating its flexibility and

effectiveness in complex scenes. Our main contributions are summarized as follows:

- We introduce the concept of flexible retrieval in the field of person re-identification for the first time and propose FlexiReID, which supports flexible retrieval with arbitrary modality combinations, opening up a new research direction.

- In FlexiReID, we propose the AEA-MoE mechanism, which dynamically selects different numbers of experts based on the input features. Additionally, we design the CMQF module, which leverages learnable embedding features to compensate for missing modalities and fuse different modality features.

- We construct a unified dataset, CIRS-PEDES, which contains four modalities. Extensive experiments demonstrate the effectiveness of FlexiReID.

## 2. Related Work

### 2.1. Vision-Language Pre-Training Models

Vision-language pre-training models, pre-trained on image-text corpora, have demonstrated significant potential in downstream vision and language tasks, such as few-shot classification(Zhou et al., 2022; Gao et al., 2024; Yu et al., 2023), cross-modality generation(Nichol et al., 2021; Ramesh et al., 2022; Patashnik et al., 2021; Niu et al., 2025b), and visual recognition(Wang et al., 2021). The pre-training approaches mainly contain the BERT-like masked-language and masked-region modeling methods(Lu et al., 2019; Su et al., 2019; Tan & Bansal, 2019; Chen et al., 2020), contrastive learning for learning a joint embedding space of vision and language(Radford et al., 2021; Jia et al., 2021; Li et al., 2021; Zhai et al., 2022a; Niu et al., 2025a), and vision-language multimodal autoregressive techniques(Cho et al., 2021; Ramesh et al., 2021). In this paper, we focus on the contrastive vision-language models (VLMs) that adopt a dual encoder to encode images and texts into the joint embedding space and use contrastive learning to align the visual and textual representations. A representative work is CLIP(Radford et al., 2021), which aggregates 400 million image-text pairs from websites. It employs a dual-encoder architecture consisting of an image encoder and a text encoder, and showcases remarkable prompt-based zero-shot performance across diverse visual classification tasks by exploiting alignments between text and image features.

### 2.2. Cross-Modal Person Re-Identification

Person re-identification (ReID) focuses on retrieving all images of a specific pedestrian across different devices, with an emphasis on learning distinctive pedestrian features. Based

on the various modalities used to represent pedestrian information, ReID can be divided into single-modal ReID(Chen et al., 2022c; Ye et al., 2021c) and cross-modal ReID(Chen et al., 2022a; Ye et al., 2021b; Zhu et al., 2021). Cross-modal ReID, in particular, addresses unique situations where RGB images of pedestrians are not readily available. It suggests utilizing non-RGB modalities(such as infrared images(Chen et al., 2022b; Yang et al., 2022; Ye et al., 2021a), text descriptions(Ding et al., 2021; Gao et al., 2021; Shao et al., 2022), and sketches(Chen et al., 2022a; Gui et al., 2020; Yang et al., 2020)) to represent pedestrian information, thereby broadening the application scope of ReID technology. Li et al.(Li et al., 2017) first propose to explore the problem of retrieving the target pedestrians with natural language descriptions for adaptation to real-world circumstances. Shao et al.(Shao et al., 2022) analyze the granularity differences between the visual modality and textual modality and propose a granularity-unified representation learning method for text-based ReID. Pang et al.(Pang et al., 2018) first propose to use professional sketches as queries to search for the target person in the RGB gallery. They design cross-domain adversarial learning methods to mine domain-invariant feature representations. In order to explore the complementarity between the sketch modality and the text modality, Zhai et al.(Zhai et al., 2022b) introduce a multi-modal ReID task that combines both sketch and text modalities as queries for retrieval. However, existing methods are limited to retrieval between specific pairs of modalities and cannot be extended to support retrieval across various combinations of modalities, which fails to meet the demands of real-world scenarios. To address this limitation, we propose the FlexiReID framework, which spans four modalities and supports seven different combinations for multimodal retrieval.

## 2.3. Mixture-of-Experts

Mixture-of-Experts (MoE) has been extensively explored in computer vision(Riquelme et al., 2021), natural language processing(Shazeer et al., 2017), and vision-language pretraining(Chen et al., 2024). MoE learns a series of expert networks and a gating network, where the outputs of the expert networks are weighted by gating scores generated by the gating network before the weighting operation. In more recent works, some researchers(Eigen et al., 2013; Shazeer et al., 2017; Fedus et al., 2022; Lepikhin et al., 2020) use the gating scores as a criterion to sparsely select only one or a few experts. The sparse activation of experts enables a significant reduction of the computational cost when training large-scale models. To further enhance computational efficiency, a TOP-K(Shazeer et al., 2017) sparse gating mechanism is employed, selecting only a subset of experts in each layer. This approach enables MoE models to scale linearly in size while maintaining manageable computational requirements, depending on the number of experts

included in the weighted averaging. In contrast to the traditional Top-K mechanism, the Adaptive Expert Activation (AEA-MOE) mechanism proposed in this paper employs an adaptive routing algorithm that dynamically adjusts the number of activated experts based on the complexity of the input data, thereby enabling more efficient utilization of computational resources.

## 3. Proposed Method

### 3.1. Framework

The overall framework is illustrated in figure2. The framework contains four types of modal data: $I_{rgb}$, $I_s$, $I_{ir}$, and $T$, corresponding to RGB, sketch, infrared, and text, respectively. We employ CLIP (ViT-B/16) as the backbone of our network. Specifically, the Image encoder is used to extract features from the three image modalities, while the Text encoder is used to extract features from the text modality.

For the Image encoder, the image is first divided into N patches, which are then mapped into embedding vectors through linear projection, with positional information added to enhance spatial awareness. Subsequently, a [CLS] token is introduced at the beginning of the embedding vectors to represent the global features of the image. These N+1 tokens are then fed into a series of transformer blocks. In order to efficiently extract features of diverse modalities, the Adaptive Expert Allocation Mixture of Experts(AEA-MOE) mechanism is introduced. After processing through the multi-head attention mechanism, an adaptive routing algorithm is employed to dynamically select the experts to be activated based on the confidence level of each expert. Unlike the traditional Top-K routing mechanism that selects a fixed number of experts, the adaptive routing algorithm can dynamically adjust the number of activated experts based on the input feature attributes. Additionally, to ensure that the adaptive routing algorithm selects the smallest necessary set of experts, an adaptive loss is introduced.

For the Text encoder, the text description T is tokenized using a simple tokenizer with a vocabulary of 49,152 words, converting it into embedding vectors $e$. A [BOS] token is added at the beginning of the sequence as a start token, and an [EOS] token is added at the end as an end token. Thus, the entire sequence can be represented as $\{e_{bos}, e_1, ..., e_{eos}\}$, and is fed into the transformer blocks. The subsequent processing is similar to that of the Image encoder and will not be repeated here.

The final image feature representations are denoted as $\{v^*_{cls}, v^*_1, ..., v^*_M\}$, and the text feature representations are denoted as $\{t_{bos}, t_1, ..., t_{eos}\}$, where * is used to indicate the three modalities within the image domain. Here, $v^*_{cls}$ represents the global features of the image, and $t_{eos}$ represents the global features of the text. Since FlexiReID

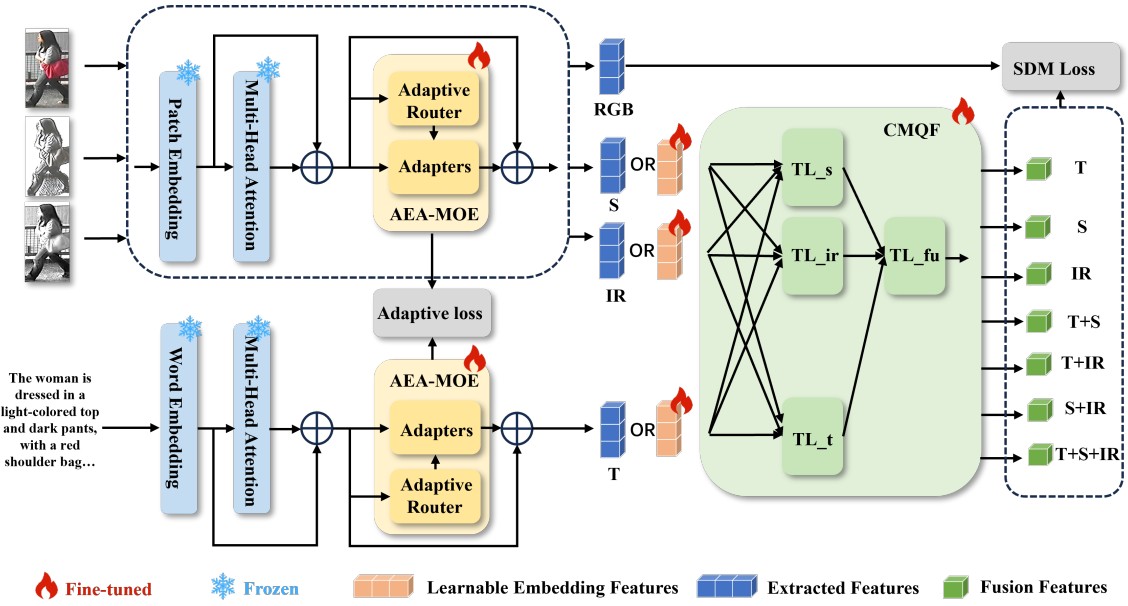

*Figure 2.* The network architecture of the proposed FlexiReID. All visual modalities share a single visual encoder. The incorporation of AEA-MOE facilitates the efficient processing of heterogeneous modality data. CMQF is employed to seamlessly integrate diverse modality features, while learnable embedding features are utilized to compensate for any missing modalities.

supports seven retrieval strategies across four modalities, we propose a feature fusion module Cross-Modal Query Fusion(CMQF), which accepts multi-modal feature input, complements the missing modalities with learnable embedding features, and finally outputs seven fused features denoted as $\{f_s, f_{ir}, f_t, f_{s\_ir}, f_{s\_t}, f_{ir\_t}, f_{s\_ir\_t}\}$. These seven fused features and visual representation $v_{cls}^{rgb}$ are finally interacted and calculated by Similarity Distribution Matching (SDM) which is an effective matching loss function across different modalities.

To reduce the number of parameters during model training, we freeze the Patch Embedding, Word Embedding, and Multi-Head Attention components of the pre-trained model, and only train the Adaptive Expert Allocation Mixture of Experts(AEA-MOE), learnable embedding features and Cross-Modal Query Fusion(CMQF) modules.

### 3.2. Adaptive Expert Allocation Mixture of Experts (AEA-MOE)

In order to enhance the multimodal feature extraction capability of the model, we introduced the AEA-MOE mechanism. The traditional MOE based on Top-K routing calculates the confidence for each expert and activates the top K experts in confidence ranking. We argue that the activation strategy overlooks the diverse characteristics inherent in the input features. Different input features require the activation of different numbers of experts, and a fixed number of experts cannot meet the needs of extracting features from

different modalities. Therefore, we propose an adaptive routing mechanism that dynamically selects the number of activated experts to extract features from different modalities. Specifically, we introduce a threshold confidence level. After the routing mechanism calculates the confidence level for each expert, it first compares the highest expert confidence level with the threshold. If the highest confidence level exceeds the threshold, only the corresponding expert is activated for feature extraction, and no additional experts are involved. If the highest confidence level does not exceed the threshold, experts are activated in descending order of confidence until the threshold is surpassed. This set of experts is then used for feature extraction. The data processing procedure of AEA-MOE can be formally represented as follows:

$$P = \text{Softmax}(W_r \cdot x^T) \tag{1}$$

$$g_i(\mathbf{x}) = \begin{cases} P_i, & Adapter_i \in S \\ 0, & Adapter_i \notin S \end{cases} \tag{2}$$

$$y = \sum_{i=1}^{n} g_i(x) Adapter_i(x) \tag{3}$$

where $W_r$ represents the weights of the gating network within the routing mechanism, and $P$ is a vector of size $n$, where $P_i$ denotes the probability of selecting the $i-$th

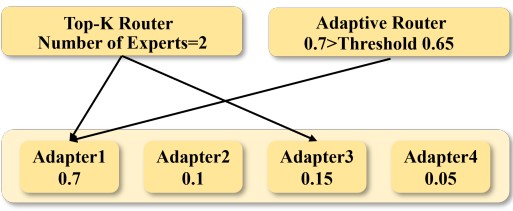

Case 1: K=2 Threshold=0.65

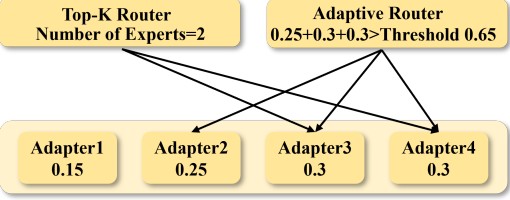

Case 2: K=2 Threshold=0.65

*Figure 3.* Traditional Top-K-based routing mechanisms are limited to selecting a fixed number of experts. In contrast, our proposed adaptive routing mechanism introduces a threshold confidence level, allowing the number of selected experts to be dynamically adjusted based on the input data.

expert as computed by the gating network. $S$ is the minimal set of experts whose confidence levels exceed the threshold. $g_i(x)$ indicates the probability of selecting the $i-$th expert as determined by the adaptive routing mechanism. Equation (3) illustrates the weighted summation process of experts and routing weights. The structure of the Adapter, inspired by (Chen et al., 2022d), consists of a down-sampling layer, an intermediate activation layer, and an up-sampling layer.

To prevent the model from assigning small weights to each expert, resulting in the need to activate a large number of experts to exceed the threshold confidence level, which contradicts the principle of MOE to enhance computational efficiency and performance by activating only a few experts. We propose an adaptive loss to constrain the distribution of the probability P calculated by the gating function. We introduce the concept of entropy for adaptive loss calculation, formally expressed as follows:

$$\mathcal{L}_{ada} = -\sum_{i=1}^{n} P_i log(P_i) \tag{4}$$

the adaptive loss ensures that the minimum necessary set of experts is activated, thereby enhancing computational efficiency. Simultaneously, it guarantees that the sum of the confidences of the activated experts exceeds the threshold confidence, and the activated experts can effectively extract features.

### 3.3. Cross-Modal Query Fusion (CMQF)

To effectively extract multimodal features and achieve flexible retrieval with various modality combinations, we propose the CMQF module. The features of the combined modalities are input into the CMQF module, and learnable embedded features are used to compensate for any missing modalities. Specifically, the features of each modality are input into their respective transformer blocks, where the query features are the sum of the query features from the other two modalities. Formally, this can be represented as:

$$y_s = TL\_s((X_{ir} + X_t)W_Q, X_sW_K, X_sW_V)$$
$$y_{ir} = TL\_ir((X_s + X_t)W_Q, X_{ir}W_K, X_{ir}W_V)$$
$$y_t = TL\_t((X_s + X_{ir})W_Q, X_tW_K, X_tW_V) \tag{5}$$

Where $X$ represents the input modality features, and $W$ denotes the weight parameters used to generate the Query, Key, and Value. $TL$ stands for the transformer block. Subsequently, all output features are concatenated and fed into a shared transformer block ($TL\_fu$). Finally, the fused features are obtained through a mean pooling operation. This process can be formally represented as follows:

$$y = \text{Concat}(y_s, y_{ir}, y_t)$$
$$f = \text{MeanPool}(TL\_fu(yW_Q, yW_K, yW_V)) \tag{6}$$

After processing through the CMQF, we obtain seven types of modality-fused features: $\{f_s, f_{ir}, f_t, f_{s\_ir}, f_{s\_t}, f_{ir\_t}, f_{s\_ir\_t}\}$. These features correspond to the following modalities: sketch, infrared, text, sketch fused with infrared, sketch fused with text, infrared fused with text, and the fusion of sketch, infrared and text features.

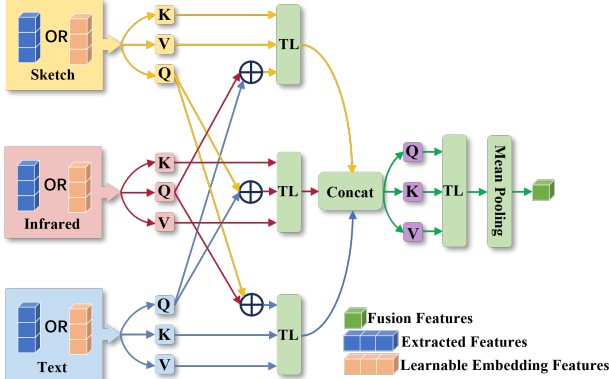

*Figure 4.* Our proposed CMQF Module.

### 3.4. Optimization and Inference

During the training phase, we utilize a parameter-free loss function called Similarity Distribution Matching

(SDM)(Jiang & Ye, 2023). Jiang and Ye incorporate the cosine similarity distributions of the $N \times N$ embeddings for image-text pairs into the KL divergence to establish a connection between the two modalities. This is formally represented as:

$$\mathcal{L}_{i2t} = KL(\mathbf{p_i}||\mathbf{q_i}) = \frac{1}{N} \sum_{i=1}^{N} \sum_{j=1}^{N} p_{i,j} \log(\frac{p_{i,j}}{q_{i,j} + \epsilon}) \quad (7)$$

$$\mathcal{L}_{sdm} = \mathcal{L}_{i2t} + \mathcal{L}_{t2i} \quad (8)$$

where $p_{i,j}$ is the probability denoting the similarity between image-text pairs and $q_{i,j}$ is the true matching probability. $\mathcal{L}_{sdm}$ is the bi-directional SDM loss.

Our model supports seven retrieval methods, each of which employs the SDM loss for its loss calculation. Therefore, the total loss is the aggregate of the SDM losses across all seven retrieval methods. Formally, this is expressed as:

$$\mathcal{L}_{sdm}^{sum} = \sum_{i=1}^{7} \mathcal{L}_{sdm}^{i} \quad (9)$$

where $\mathcal{L}_{sdm}^{i}$ denotes the SDM loss for the $i\text{-}th$ retrieval method. Our framework is trained end-to-end, and the overall optimization objective is formally defined as:

$$\mathcal{L} = \mathcal{L}_{sdm}^{sum} + \lambda \mathcal{L}_{ada} \quad (10)$$

During inference, the trained network processes combinations of various modalities, assigning learnable Embedding Features to any missing modalities. It subsequently extracts integrated features across these modalities and computes their similarity with the RGB image embeddings. The Top-K candidates are then processed to derive the relevant evaluation metrics for each query.

## 4. Experiments

### 4.1. Experimental Setup

**Datasets.** In this study, we introduced four datasets: CUHK-PEDES, ICFG-PEDS, RSTPReid, and SYSU-MM01. To support the flexible retrieval capabilities of FlexiReid, which accommodates four modalities of pedestrian data, we expanded the modalities of these datasets. For the CUHK-PEDES, ICFG-PEDS, and RSTPReid datasets, which originally contain RGB images and textual descriptions, we employed StyleGAN3(Karras et al., 2021) to generate sketch modalities. This involved preprocessing each RGB image and using a pre-trained StyleGAN3 model to convert it into a sketch, preserving the primary contours and structural information of the original image. Concurrently, we used InfraGAN(Özkanoğlu & Ozer, 2022) to generate infrared

image modalities. This was achieved by training the model on pairs of visible light and corresponding infrared images, and then applying the trained model to each RGB image to produce infrared images that capture thermal radiation information. For the SYSU-MM01 dataset, which originally contains RGB and infrared images, we similarly used Style-GAN3(Karras et al., 2021) to generate sketch modalities and employed the GPT-4 model to generate textual descriptions. This involved extracting features from each RGB image and generating corresponding textual descriptions that capture the main content and characteristics of the images. By incorporating these additional modalities, our expanded datasets better simulate real-world multimodal data scenarios, providing richer and more diverse data support for the training of the FlexiReid model. An overview of training and test set partitioning for each dataset can be found in the existing work(Ding et al., 2021; Zhu et al., 2021; Wu et al., 2020).

**Evaluation Protocols.** Following existing cross-modality ReID settings(Chen et al., 2022a; Ye et al., 2021b;c), we use the Rank-k matching accuracy, mean Average Precision (mAP), and mean Inverse Negative Penalty (mINP)(Ye et al., 2021c) metrics for performance evaluation in our FlexiReID.

**Implementation Details.** We employ the Vision Transformer(Dosovitskiy, 2020) as the visual modalities feature learning backbone, and the Transformer model(Vaswani, 2017) as the textual modality feature learning backbone. Both backbones have pre-trained parameters derived from CLIP(Radford et al., 2021). During the training process, all parameters of the backbone networks are frozen. In a batch, we randomly select 64 identities, each containing a sketch, an infrared, a text, and an RGB sample. The image is resized to $384 \times 128$, and the length of textual token sequence is 77. We train our FlexiReID model with the Adam optimizer for 60 epochs. And the initial learning rate is computed as 1e-5 and decayed by a cosine schedule. The threshold confidence level is set to 0.6, and the number of experts is 6. The hyperparameter $\lambda$ that indicates the adaptive loss is set to 0.5. We perform experiments on a single NVIDIA 3090 24GB GPU.

### 4.2. Performance Comparison

**Results on CUHK-PEDES, ICFG-PEDES and RST-PReid** In our experiments, we evaluated the model's performance across seven different test tasks: $T \rightarrow R$, $S \rightarrow R$, $IR \rightarrow R$, $T + S \rightarrow R$, $T + IR \rightarrow R$, $S + IR \rightarrow R$, and $T + S + IR \rightarrow R$ The results, as shown in table 1, demonstrate the model's performance on the CHUK-PEDES, ICFG-PEDES, and RSTPReid datasets. In the $T \rightarrow R$ task, our model achieved a Rank-K accuracy of 69.20% at R@1, 86.43% at R@5, and 91.41% at R@10 on the CHUK-PEDES dataset. On the ICFG-PEDES dataset, the model

*Table 1.* Comparison with the state-of-the-arts on CUHK-PEDES, ICFG-PEDES, and RSTPReid datasets. Rank ($R$) at $k$ accuracy (%) is reported. The best results are bold.

| Tasks | Methods | Venue | CUHK-PEDES | | | | | ICFG-PEDES | | | | | RSTPReid | | | | |
|---|---|---|---|---|---|---|---|---|---|---|---|---|---|---|---|---|---|
| | | | R1 | R5 | R10 | mAP | mINP | R1 | R5 | R10 | mAP | mINP | R1 | R5 | R10 | mAP | mINP |
| T→R | CMPM/C(Zhang & Lu, 2018) | ECCV18 | 49.37 | 71.69 | 79.27 | - | - | 43.51 | 65.44 | 74.26 | - | - | - | - | - | - | - |
| | MIA(Niu et al., 2020) | TIP20 | - | - | - | - | - | 46.49 | 67.14 | 75.18 | - | - | - | - | - | - | - |
| | ViTAA(Wang et al., 2020) | ECCV20 | 55.97 | 75.84 | 83.52 | - | - | 50.98 | 68.79 | 75.78 | - | - | - | - | - | - | - |
| | NAFS(Gao et al., 2021) | arXiv21 | 59.36 | 79.13 | 86.00 | 54.07 | - | - | - | - | - | - | - | - | - | - | - |
| | DSSL(Zhu et al., 2021) | MM21 | 59.98 | 80.41 | 87.56 | - | - | - | - | - | - | - | 32.43 | 55.08 | 63.19 | - | - |
| | SSAN(Ding et al., 2021) | arXiv21 | 61.37 | 80.15 | 86.73 | - | - | 54.23 | 72.63 | 79.53 | - | - | 43.50 | 67.80 | 77.15 | - | - |
| | Han et al.(Han et al., 2021) | BMVC21 | 64.08 | 81.73 | 88.19 | 60.08 | - | - | - | - | - | - | - | - | - | - | - |
| | LBUL+BERT(Wang et al., 2022b) | MM22 | 64.04 | 82.66 | 87.22 | - | - | - | - | - | - | - | 45.55 | 68.20 | 77.85 | - | - |
| | SAF(Li et al., 2022) | ICASSP22 | 64.13 | 82.62 | 88.40 | 58.61 | - | 54.86 | 72.13 | 79.13 | 32.76 | - | 44.05 | 67.30 | 76.25 | 36.81 | - |
| | TIPCB(Chen et al., 2022e) | Neuro22 | 64.26 | 83.19 | 89.10 | - | - | 54.96 | 74.72 | 81.89 | - | - | - | - | - | - | - |
| | CAIBC(Wang et al., 2022a) | MM22 | 64.43 | 82.87 | 87.35 | - | - | - | - | - | - | - | 47.35 | 69.55 | 79.00 | - | - |
| | AXM-Net(Farooq et al., 2022) | MM22 | 64.44 | 80.52 | 86.77 | 58.73 | - | - | - | - | - | - | - | - | - | - | - |
| | LGUR(Shao et al., 2022) | MM22 | 65.25 | 83.12 | 89.00 | - | - | 59.02 | 75.32 | 81.56 | - | - | - | - | - | - | - |
| | IVT(Shu et al., 2022) | ECCV22 | 65.59 | 83.11 | 89.21 | - | - | 56.04 | 73.60 | 80.22 | - | - | 46.70 | 70.00 | 78.80 | - | - |
| | UNIReID(Chen et al., 2023a) | CVPR23 | 68.71 | 85.35 | 90.84 | - | - | 61.28 | 77.40 | 83.16 | - | - | 60.25 | 79.85 | 87.10 | - | - |
| | CFine(Yan et al., 2023) | TIP23 | 69.57 | 85.93 | 91.15 | - | - | 60.83 | 76.55 | 82.42 | - | - | 50.55 | 72.50 | 81.60 | - | - |
| | CSKT(Liu et al., 2024b) | ICASSP24 | 69.70 | 86.92 | 91.80 | 62.74 | - | 58.90 | 77.31 | 83.56 | 33.87 | - | 57.75 | 81.30 | 88.35 | 46.43 | - |
| | FlexiReID(Ours) | - | 69.20 | 86.43 | 91.41 | 62.47 | 48.32 | 61.34 | 78.41 | 83.92 | 35.73 | 7.53 | 55.79 | 79.62 | 86.48 | 45.37 | 26.25 |
| S→R | Sketch Trans+(Chen et al., 2023b) | PAMI23 | 81.39 | 90.61 | 93.54 | 73.72 | 64.72 | 74.83 | 86.75 | 91.52 | 38.64 | 5.68 | 61.37 | 80.15 | 88.29 | 48.94 | 25.73 |
| | DALNet(Liu et al., 2024a) | AAAI2024 | 83.03 | 92.39 | 94.58 | 75.39 | 66.82 | 77.28 | 87.84 | 92.61 | 40.35 | 6.18 | 64.68 | 83.27 | 89.06 | 51.08 | 27.13 |
| | UNIReID(Chen et al., 2023a) | CVPR23 | 84.87 | - | - | 78.85 | 68.55 | 77.47 | - | - | 40.41 | 6.31 | 65.80 | - | - | 51.22 | 27.47 |
| | FlexiReID(Ours) | - | 84.92 | 93.17 | 95.02 | 79.21 | 68.83 | 79.28 | 89.69 | 93.37 | 41.21 | 6.85 | 66.79 | 84.52 | 90.39 | 52.72 | 28.36 |
| T+S→R | UNIReID(Chen et al., 2023a) | CVPR23 | 86.29 | - | - | 80.92 | 71.30 | 82.17 | - | - | 47.00 | 8.74 | 73.20 | - | - | 58.72 | 34.61 |
| | FlexiReID(Ours) | - | 87.47 | 94.51 | 96.14 | 82.43 | 72.96 | 83.82 | 93.49 | 95.63 | 47.81 | 9.03 | 76.10 | 89.74 | 94.31 | 64.73 | 41.24 |
| IR→R | GUR(Yang et al., 2023) | ICCV23 | 82.06 | 91.72 | 93.95 | 75.84 | 66.86 | 80.31 | 90.89 | 92.78 | 44.36 | 6.90 | 73.42 | 86.29 | 91.35 | 60.43 | 38.52 |
| | SDCL(Yang et al., 2024) | CVPR24 | 84.57 | 92.73 | 94.58 | 77.32 | 68.20 | 81.36 | 91.83 | 94.07 | 45.81 | 7.92 | 74.67 | 87.94 | 93.16 | 62.75 | 39.93 |
| | FlexiReID(Ours) | - | 85.26 | 93.25 | 95.31 | 79.43 | 69.39 | 82.03 | 92.19 | 94.27 | 46.76 | 8.47 | 75.36 | 88.71 | 93.27 | 63.22 | 40.82 |
| T+IR→R | FlexiReID(Ours) | - | 86.23 | 94.07 | 96.52 | 81.49 | 70.96 | 82.42 | 92.56 | 95.08 | 47.35 | 8.75 | 75.84 | 89.28 | 94.11 | 63.90 | 40.98 |
| S+IR→R | FlexiReID(Ours) | - | 85.97 | 93.52 | 95.89 | 81.02 | 69.68 | 82.57 | 92.74 | 95.28 | 47.43 | 8.93 | 75.94 | 89.47 | 94.23 | 64.59 | 41.18 |
| T+S+IR→R | FlexiReID(Ours) | - | **88.23** | **95.13** | **96.75** | **82.63** | **73.05** | **84.26** | **93.78** | **96.15** | **48.09** | **9.42** | **76.35** | **90.26** | **95.08** | **65.19** | **42.29** |

recorded a Rank-K accuracy of 61.34% at R@1, 78.41% at R@5, and 83.92% at R@10. For the RSTPReid dataset, the Rank-K accuracy was 55.79% at R@1, 79.62% at R@5, and 86.48% at R@10. Notably, the model achieved state-of-the-art (SOTA) performance on the ICFG-PEDES dataset and demonstrated performance close to SOTA on the CHUK-PEDES and RSTPReid datasets. These results underscore the model's robust capability in text-to-image retrieval tasks. In the S→R task, our model achieved MAP metrics of 79.21%, 41.21%, and 52.72% on the three datasets, respectively. These results represent state-of-the-art (SOTA) performance across all datasets.

In other tasks ($T + S \rightarrow R$, $T + IR \rightarrow R$, $S + IR \rightarrow R$, $T + S + IR \rightarrow R$), our model also demonstrated robust performance. Compared to single-modality retrieval methods, the flexible combination of different modalities consistently achieved superior results. For instance, in the T+S+IR→R task, our model achieved Rank-K metrics of 88.23% at R@1, 95.13% at R@5, and 96.75% at R@10 on the CUHK-PEDES dataset, surpassing single-modality retrieval methods. This indicates that combining various modalities provides richer pedestrian detail information, thereby enhancing overall retrieval performance.

Overall, most current cross-modal ReID methods focus on retrieving RGB modality from a non-RGB modality. Our approach not only achieves state-of-the-art (SOTA) or near-SOTA performance in single-modality retrieval tasks but also supports a broader range of retrieval methods. The flexible combination of various modalities outperforms single-modality cross-modal retrieval methods, making our approach more versatile and widely applicable.

*Table 2.* Comparison with the state-of-the-arts on SYSU-MM01 datasets. Rank ($R$) at $k$ accuracy (%) is reported. The best results are bold.

| Tasks | Methods | Venue | All-Search | | | | Indoor-Search | | | |
|---|---|---|---|---|---|---|---|---|---|---|
| | | | R1 | R10 | R20 | mAP | R1 | R10 | R20 | mAP |
| IR→R | SSFT(Lu et al., 2020) | CVPR20 | 61.6 | 89.2 | 93.9 | 63.3 | 70.5 | 94.9 | 97.7 | 72.6 |
| | DDAG(Ye et al., 2020) | ECCV20 | 54.8 | 90.4 | 95.8 | 53.0 | 61.0 | 94.1 | 98.4 | 68.0 |
| | DG-VAE(Pu et al., 2020) | MM20 | 59.5 | 93.8 | - | 58.5 | - | - | - | - |
| | CICL+IAMA(Zhao et al., 2021) | AAAI21 | 57.2 | 94.3 | 98.4 | 59.3 | 66.6 | 98.8 | 99.7 | 74.7 |
| | VCD+VML(Tian et al., 2021) | CVPR21 | 60.0 | 94.2 | 98.1 | 58.8 | 66.1 | 96.6 | 99.4 | 73.0 |
| | MPANet(Wu et al., 2021) | CVPR21 | 70.6 | 96.2 | 98.8 | 68.2 | 76.7 | 98.2 | 99.6 | 81.0 |
| | MCLNet(Hao et al., 2021) | ICCV21 | 65.4 | 93.3 | 97.1 | 62.0 | 72.6 | 97.0 | 99.2 | 76.6 |
| | SMCL(Wei et al., 2021) | ICCV21 | 67.4 | 92.9 | 96.8 | 61.8 | 68.8 | 96.6 | 98.8 | 75.6 |
| | FlexiReID(Ours) | - | 67.9 | 93.4 | 97.6 | 62.5 | 69.2 | 97.2 | 99.1 | 75.8 |
| T→R | UNIReID(Chen et al., 2023a) | CVPR23 | 54.3 | 90.2 | 95.7 | 63.8 | 56.7 | 91.8 | 96.5 | 66.9 |
| | FlexiReID(Ours) | - | 56.8 | 92.7 | 96.3 | 65.4 | 58.2 | 93.3 | 97.4 | 67.6 |
| S→R | UNIReID(Chen et al., 2023a) | CVPR23 | 64.7 | 90.9 | 94.6 | 59.2 | 66.7 | 95.4 | 97.8 | 74.0 |
| | FlexiReID(Ours) | - | 66.4 | 92.7 | 95.2 | 60.3 | 68.5 | 96.7 | 98.2 | 75.3 |
| T+S→R | UNIReID(Chen et al., 2023a) | CVPR23 | 66.5 | 94.2 | 97.9 | 66.4 | 69.3 | 94.6 | 97.5 | 72.8 |
| | FlexiReID(Ours) | - | 68.7 | 95.1 | 98.3 | 67.2 | 70.6 | 95.0 | 97.9 | 73.4 |
| T+IR→R | FlexiReID(Ours) | - | 68.9 | 96.7 | 98.2 | 68.1 | 71.2 | 98.3 | 99.3 | 75.2 |
| S+IR→R | FlexiReID(Ours) | - | 69.0 | 96.4 | 98.5 | 68.6 | 72.2 | 98.7 | 99.4 | 76.2 |
| T+S+IR→R | FlexiReID(Ours) | - | **71.3** | **97.2** | **98.7** | **69.5** | **73.2** | **98.9** | **99.6** | **77.3** |

**Results on SYSU-MM01** We also evaluated our model on the SYSU-MM01 dataset, as shown in table 2. In the IR → RGB task under the full search mode, our model demonstrated competitive performance, approaching the state-of-the-art (SOTA) level. However, by flexibly combining other modalities, performance can be further enhanced. For instance, in the Indoor-Search mode for the $T + S + IR \rightarrow R$ task, our model achieved Rank-K metrics of 73.2% on R@1, 98.9% on R@10, and 99.6% on R@20. This improvement is attributed to the introduction of text and sketch modalities, which provide additional pedestrian features and enhance the model's comprehension capabilities.

*Table 3.* Ablation study on R@1 about each component of FlexiReID. The metric Avg. denotes the average R@1 across seven search modes.

| No. | Module | Components | | | | | $T \to R$ | $S \to R$ | $IR \to R$ | $T + S \to R$ | $T + IR \to R$ | $S + IR \to R$ | $T + S + IR \to R$ | Avg. |
|---|---|---|---|---|---|---|---|---|---|---|---|---|---|---|
| | | MLP-Adapter | AEA-MOE | AL | CMQF | LEF | | | | | | | | |
| 0 | Zero-shot CLIP | | | | | | 12.58 | 2.47 | 3.79 | 1.62 | 2.73 | 3.72 | 4.51 | 4.49 |
| 1 | +MLP-Adapter | ✓ | | | | | 66.38 | 81.47 | 81.93 | 83.58 | 83.26 | 82.43 | 85.03 | 80.58 |
| 2 | +AEA-MOE(w/o AL) | ✓ | ✓ | | | | 68.45 | 83.08 | 83.74 | 85.83 | 85.52 | 84.32 | 86.71 | 82.52 |
| 3 | +AEA-MOE(w/ AL) | ✓ | ✓ | ✓ | | | 68.87 | 84.13 | 84.37 | 86.41 | 85.93 | 84.97 | 87.35 | 83.14 |
| 4 | +CMQF(w/o LEF) | ✓ | ✓ | ✓ | ✓ | | 69.08 | 84.59 | 85.12 | 87.38 | 86.15 | 85.78 | 88.05 | 83.73 |
| 5 | +CMQF(w/ LEF) | ✓ | ✓ | ✓ | ✓ | ✓ | 69.20 | 84.92 | 85.26 | 87.47 | 86.23 | 85.97 | 88.23 | 83.90 |

## 4.3. Ablation Study

A comprehensive ablation study for components of FlexiReID is presented in Table 3, including the most critical accuracy metric R@1 and the average metric on CUHK-PEDES datasets. The results in No.0 serve as the backbone baseline by zero-shot CLIP, where inference is performed directly on the original frozen CLIP model without adding any additional trainable modules. No.1 employs the traditional MOE method, while No.2 and No.3 utilize AEA-MOE. It is evident that AEA-MOE outperforms the traditional MOE in handling data from various modalities, thereby achieving superior performance. Furthermore, incorporating adaptive loss further enhances performance. As demonstrated by No.4, CMQF plays a pivotal role in the integration of features from different modalities, thereby improving the model's performance. Moreover, substituting missing modalities with Learnable Embedding Features yields optimal performance. These results indicate that each component of FlexiReID significantly contributes to the model's overall performance, working in concert to achieve optimal outcomes.

**Ablation Study on Routing Strategies** To evaluate the effectiveness of our proposed adaptive routing mechanism, we conduct a comprehensive ablation study by comparing it with several widely adopted routing strategies, including Top-K routing, soft routing, and hash routing. As shown in Table 4, our adaptive routing achieves the best overall performance. These results demonstrate that our method benefits from dynamically selecting both the number and combination of experts based on the input features, rather than relying on fixed or probabilistic routing strategies. This ablation study highlights the critical role of adaptive expert allocation in enhancing feature expressiveness and improving downstream retrieval performance.

*Table 4.* Comparison of different routing strategies.

| Method | Avg. |
|---|---|
| Top-K Routing | 80.58 |
| Soft Routing | 81.80 |
| Hash Routing | 83.11 |
| Ours (Adaptive Routing) | **83.90** |

**Ablation Study on Feature Fusion Strategies** To validate the effectiveness of the proposed Cross-Modal Query Fusion (CMQF) module, we conducted an ablation study comparing it with several commonly used multimodal feature fusion strategies, including concatenation, summation, and hierarchical fusion. Specifically, the concatenation strategy directly concatenates features from different modalities and feeds them into a shared Transformer for fusion. The summation strategy aggregates features by summing them prior to Transformer processing. The hierarchical fusion approach first encodes each modality independently using separate Transformers and then performs fusion at a shared layer. All methods were evaluated on the modality-augmented CUHK-PEDES dataset, and the average R@1 accuracy across seven retrieval tasks was reported. As shown in Table 5, CMQF achieved the best overall performance. Compared to the aforementioned methods, CMQF leverages cross-modal attention mechanisms and incorporates learnable embedding features to compensate for missing modalities, enabling more comprehensive and robust feature alignment. These results demonstrate the superior capability of CMQF in enhancing multimodal feature representations and improving retrieval accuracy.

*Table 5.* Comparison of different feature fusion strategies

| Method | Avg. |
|---|---|
| Concatenation | 83.14 |
| Summation | 82.24 |
| Hierarchical Fusion | 83.51 |
| CMQF (Ours) | **83.90** |

## 4.4. Hyper-Parameter Analysis

**The Number of Experts.** As shown in Figure 5a, to investigate the impact of the number of experts n, we sample n as 2, 4, 6, 8 and 10, to evaluate the R@1 accuracy and mAP under different numbers of experts. A constant threshold confidence level is fixed to 0.4 in the overall experiment. When n is less than 6, average R@1 performance gradually increases with an increasing number of experts. However, when n exceeds 6, larger n leads to a decrease in performance. we observe that although increasing n can proportionally enhance the model's information capacity, a larger n does not necessarily lead to better performance. This suggests that the model's capacity cannot grow indefinitely. We ultimately determine n = 6 as a practical choice.

**Threshold Confidence Level** As shown in Figure 5b, we set the number of experts n = 6 and explore the R@1 accuracy and mAP with the change of threshold confidence level. When the confidence level is low, the model's performance is poor. However, as the threshold confidence level increases to 0.4, the performance reaches its peak. As the threshold confidence level increases further, the model's performance essentially reaches a plateau without additional improvement. Therefore, the threshold confidence level is set to 0.4.

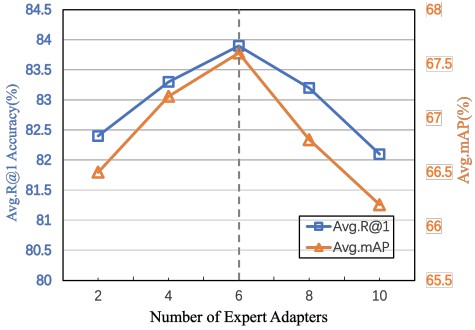

(a) Ablation Study on the Number of Experts.

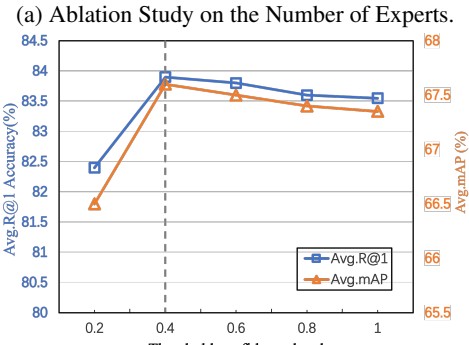

(b) Ablation Study on Threshold Confidence.

*Figure 5.* Ablation Study on the Number of Experts and Threshold Confidence.

## 5. Conclusion

In this paper, we propose FlexiReID, a multimodal person re-identification framework that supports flexible retrieval across four modalities: text, sketches, RGB, and infrared, as well as any combination thereof. We design the AEA-MoE mechanism to dynamically select expert networks and introduce the CMQF module to optimize cross-modal feature fusion. Based on the expanded CIRS-PEDES dataset, experimental results demonstrate that FlexiReID outperforms existing methods in complex scenarios, validating its flexibility and effectiveness, and opening up new research directions for multimodal person re-identification.

## Acknowledgements

This work was supported by the National Science Fund for Distinguished Young Scholars (No.62025603), the National Natural Science Foundation of China (No. U21B2037, No. U22B2051, No. U23A20383, No. U21A20472, No. 62176222, No. 62176223, No. 62176226, No. 62072386, No. 62072387, No. 62072389, No. 62002305 and No. 62272401), and the Natural Science Foundation of Fujian Province of China (No. 2021J06003, No.2022J06001).

## Impact Statement

This paper presents work whose goal is to advance the field of Machine Learning. There are many potential societal consequences of our work, none which we feel must be specifically highlighted here.

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

## A. Ablation Study on the Impact of Applying Adaptive Moe to Different Encoders.

To further investigate the applicability and effectiveness of our proposed AEA-MOE mechanism within the FlexiReID framework, we design an ablation study presented in the appendix. In this study, the AEA-MOE module is applied individually to the image encoder, the text encoder, and jointly to both, aiming to clarify the performance contribution of this mechanism across different modality-specific processing modules. As shown in the table 6, Experiment No.1 incorporates AEA-MOE only into the image encoder, No.2 applies it to the text encoder, and No.3 integrates it into both encoders. Based on the Rank@1 accuracy across seven retrieval tasks, we observe that AEA-MOE improves performance in both the image and text encoders. However, the enhancement is more significant on the image side, and the best overall performance is achieved when applied to both encoders simultaneously.

These results validate the generalizability of the proposed AEA-MOE module. It can be effectively adapted to both image and text modalities and demonstrates a complementary enhancement effect in multimodal retrieval scenarios.

*Table 6.* Ablation results on the CUHK-PEDES dataset evaluating the impact of applying the AEA-MOE mechanism to the image encoder, text encoder, or both.

| No. | Method | $T \rightarrow R$ | $S \rightarrow R$ | $IR \rightarrow R$ | $T+S \rightarrow R$ | $T+IR \rightarrow R$ | $S+IR \rightarrow R$ | $T+S+IR \rightarrow R$ | **Avg.** |
|---|---|---|---|---|---|---|---|---|---|
| 0 | MLP Adapter | 66.38 | 81.47 | 81.93 | 83.58 | 83.26 | 82.43 | 85.03 | 80.58 |
| 1 | No.0+AEA-MOE(Image) | 68.24 | 83.73 | 83.81 | 85.87 | 85.39 | 84.46 | 86.88 | 82.63 |
| 2 | No.0+AEA-MOE(Text) | 67.25 | 82.66 | 82.78 | 84.52 | 84.06 | 83.19 | 85.74 | 81.46 |
| 3 | No.0+AEA-MOE(Image,Text) | 68.87 | 84.13 | 84.37 | 86.41 | 85.93 | 84.97 | 87.35 | 83.14 |

## B. Experiments on PKU-Sketch

To further validate the effectiveness of the FlexiReID framework in handling sketch-based scenarios, we conduct additional experiments on the PKU-Sketch dataset, which consists of two modalities: sketches and RGB images. To address the issue of missing modalities, we adopt a Learnable Embedding Features (LEF) strategy, wherein each missing modality is compensated by a corresponding learnable embedding vector. These embedding vectors are jointly optimized with the main network parameters during training, enabling semantic completion and consistency preservation for the missing modalities. This allows the model to perform robust cross-modal retrieval even when the input modality combination is incomplete.

As shown in the figure 7, FlexiReID achieves superior performance compared to existing methods on the PKU-Sketch dataset. This demonstrates that FlexiReID exhibits strong robustness and practical applicability in complex scenarios with incomplete multimodal inputs.

*Table 7.* Sketch-based retrieval performance on the PKU-Sketch dataset.

| Method | Reference | mAP | Rank@1 | Rank@5 | Rank@10 |
|---|---|---|---|---|---|
| CCSC | MM22 | 83.7 | 86.0 | 98.0 | 100.0 |
| Sketch Trans+ | PAMI2023 | - | 85.8 | 96.0 | 99.0 |
| DALNet | AAAI2024 | 86.2 | 90.0 | 98.6 | 100.0 |
| FlexiReID(Ours) | - | 91.2 | 93.5 | 99.3 | 100.0 |

## C. Experiments on Market-1501 and MSMT17

To further evaluate the performance of the FlexiReID framework on the traditional RGB-to-RGB single-modal person re-identification task, we conduct experiments on two widely used ReID datasets: Market-1501 and MSMT17. This experiment aims to verify whether our proposed framework can still exhibit strong visual feature modeling capabilities without relying on auxiliary modality information. Notably, to maintain architectural consistency with multimodal retrieval settings, we utilize learnable embedding features to replace the missing modalities. This mechanism ensures that even when only RGB images are available, the model can still receive a unified four-modality representation, thereby preserving the semantic generalization ability inherent in the multimodal design.

As shown in the table 8, FlexiReID outperforms existing representative methods on both Market-1501 and MSMT17. These results clearly demonstrate that our method not only excels in complex cross-modal retrieval scenarios but also delivers outstanding performance in standard RGB-RGB settings, highlighting its strong generalizability and practical applicability.

*Table 8.* Evaluation results on the Market-1501 and MSMT17 datasets for the RGB-to-RGB person re-identification task.

| Methods | Reference | Market-1501 | | MSMT17 | |
|---------|-----------|-------------|-----|--------|-----|
| | | Rank@1 | mAP | Rank@1 | mAP |
| FastReID | ACM MM23 | 95.4 | 88.2 | 83.3 | 59.9 |
| BPBreID | WACV23 | 95.1 | 87.0 | - | - |
| MVI2P | Inf Fusion24 | 95.2 | 87.0 | 80.4 | 56.4 |
| FlexiReID(Ours) | - | 96.0 | 92.1 | 83.7 | 67.5 |

