# OpenReview forum: "FlexiReID: Adaptive Mixture of Expert for Multi-Modal Person Re-Identification"
_ICML.cc/2025/Conference — ICML 2025 poster_

### Official Review · Reviewer_33pg · 2025-03-07

**Overall Recommendation:** 4

**Summary:**

This article innovatively introduces the concept of combined modality pedestrian re-identification. Compared with traditional cross-modal identification, it demonstrates greater flexibility in dealing with complex scenarios. Centered around this concept, the article constructs the FlexiReID framework, which integrates a variety of cross-modal and combined modality pedestrian identification tasks, and exhibits extremely strong universality, enabling it to adapt well to diverse application scenarios.

**Claims And Evidence:**

This work's claims are intuitive, convincing, and supported by its experiments.

**Essential References Not Discussed:**

In the related work section, the author could include a review of the literature on feature fusion.

**Experimental Designs Or Analyses:**

Overall, the experimental design in this paper demonstrates a high level of rationality. The study selects datasets that encompass diverse scene characteristics and conducts comprehensive evaluations on a variety of ReID tasks. The experimental results clearly indicate that the proposed method exhibits significant advantages in key performance metrics such as accuracy and recall. Furthermore, through carefully designed ablation studies, the paper convincingly validates that multimodal collaborative processing achieves superior recognition accuracy compared to unimodal independent operation, highlighting the powerful efficacy of multimodal fusion.

**Methods And Evaluation Criteria:**

The FlexiReID framework proposed in this paper is highly forward-looking. It breaks away from the traditional single-task mode and constructs a comprehensive and multi-level recognition system by integrating various identification tasks. In practical application scenarios, pedestrian data features are rich and diverse. Single cross-modal recognition is difficult to comprehensively capture key information. However, the combined modality recognition advocated by FlexiReID can fully explore the complementarity among different modality data, greatly improving the recognition accuracy. The fusion of multiple modality data makes the recognition more accurate. This innovative concept not only meets the complex real-world needs but also opens up a new path for the widespread application of ReID technology. It is expected to become the core direction of future industry development.

**Other Comments Or Suggestions:**

The FlexiReID framework proposed in this paper is truly distinctive in the field of cross-modal person re-identification (ReID), offering a novel perspective for addressing the challenges of ReID in complex scenarios.

However, in the experimental validation section, while the paper conducts ablation studies to verify the effectiveness of each module within the overall framework, the specific role of the adaptive MoE mechanism in processing different modalities could be further explored. For instance, additional ablation studies could be conducted to independently analyze the impact of applying the adaptive MoE mechanism on the text and image modalities, providing deeper insights into its contributions to overall performance.

**Other Strengths And Weaknesses:**

Strengths：

- The research findings of this paper are not limited to the field of cross-modal person re-identification. For instance, its proposed concept of modality combination and adaptive mechanisms provide new insights and methodological references for addressing challenges in complex data fusion and dynamic model adjustment across various related domains.

Weaknesses：

- The authors could conduct additional ablation studies to independently analyze the impact of applying the adaptive MoE mechanism to the text and image modalities on performance.

- In the related work section, the authors could further supplement the discussion with a comprehensive review and literature survey on feature fusion research.

**Questions For Authors:**

- In the feature fusion module, have you considered directly fusing two modalities without using learnable features as replacements when a modality is missing?

- In the ablation experiment section, how is feature fusion performed when the feature fusion module is not used?

**Relation To Broader Scientific Literature:**

The FlexiReID framework proposed in this paper expands the research paradigm of cross-modal ReID, extending from traditional single-modal queries (such as text-to-image or infrared-to-image) to multi-modal combination retrieval, an aspect that has not been fully explored in existing literature. Previous studies mainly focused on cross-modal feature alignment and modality-invariant feature learning, such as modality transformation methods based on adversarial learning or Transformers, but none considered the possibility of multi-modal joint queries. Outrageously Large Neural Networks: The Sparsely-Gated Mixture-of-Experts Layer first introduced the MOE model based on the Top-K routing mechanism. In contrast, this paper proposes an adaptive MoE mechanism, which is more dynamic than the traditional Top-K MoE method and can flexibly select the number of active experts. Additionally, the feature fusion method designed in the paper, combined with a learnable feature filling strategy, effectively enhances retrieval performance in the case of missing modalities, which is uncommon in previous research based on simple feature concatenation or shared representation learning. Overall, I believe FlexiReID is a more flexible, universal, and efficient solution for person re-identification.

**Theoretical Claims:**

The theory proposed in this paper is highly compelling and well-supported by solid evidence. Taking the attention allocation mechanism in the model as an example, it innovatively introduces a dynamic weighting strategy based on the distribution of data features. This strategy adaptively adjusts the weight proportions of submodules according to the degree of dispersion in the input data features. Furthermore, in the feature fusion stage, the paper adopts a progressive fusion architecture. Initially, features from different modalities are processed separately, followed by cross-mapping and fusion between features, gradually integrating information across modalities. This hierarchical, multi-stage fusion approach effectively captures the correlations between different modality features.

---

> ### Author Rebuttal · Authors · 2025-03-31
>
> > In the related work section, the author could include a review of the literature on feature fusion.
> >
>
> A1: Thank you for the suggestion. We appreciate your insight and will include a review of relevant literature on feature fusion in the related work section in the revised version.
>
> > The authors could conduct additional ablation studies to independently analyze the impact of applying the adaptive MoE mechanism to the text and image modalities on performance.
> >
>
> A2: Thank you for your suggestion. As requested, we have added the corresponding ablation experiments, with the results shown in the table below. In Row No.1, the adaptive MoE mechanism is applied on the image modality side, while in Row No.2, it is applied on the text modality side. As shown, applying the adaptive MoE to the image modality yields a greater performance improvement. This is because the image branch contains three modalities, making it more suitable for the adaptive MoE mechanism.
>
> Talbe1: Ablation study on the impact of applying adaptive moe to different encoders
>
> | No. | Method | T—R | S—R | IR—R | T+S—R | T+IR—R | S+IR—R | T+S+IR—R | Avg. |
> | --- | --- | --- | --- | --- | --- | --- | --- | --- | --- |
> | No.0 | MLP_Adapter | 66.38 | 81.47 | 81.93 | 83.58 | 83.26 | 82.43 | 85.03 | 80.58 |
> | No.1 | No.0+AEA-MOE(Image) | 68.24 | 83.73 | 83.81 | 85.87 | 85.39 | 84.46 | 86.88 | 82.63 |
> | No.2 | No.0+AEA-MOE(Text) | 67.25 | 82.66 | 82.78 | 84.52 | 84.06 | 83.19 | 85.74 | 81.46 |
> | No.3 | No.0+AEA-MOE(Image,Text) | 68.87 | 84.13 | 84.37 | 86.41 | 85.93 | 84.97 | 87.35 | 83.14 |
>
> > In the feature fusion module, have you considered directly fusing two modalities without using learnable features as replacements when a modality is missing?
> >
>
> A3: Thank you for your question. In fact, as shown in Row No.4 of the ablation study in Table 3 of the paper, we conducted an experiment without using learnable features to replace the missing modalities. The comparison shows that using learnable features to fill in missing modalities is beneficial for performance improvement.
>
> > In the ablation experiment section, how is feature fusion performed when the feature fusion module is not used?
> >
>
> A4: Thank you for your question. In the ablation study, when the feature fusion module is not used, we concatenate the features and pass them through a Transformer module to extract global features for fusion. This corresponds to the fusion method in Row No.1 of Table 1 in our response to reviewer NWLb. Additionally, the same table includes ablation experiments comparing different fusion strategies, which clearly demonstrate the performance advantages of our CMQF method.

---

### Official Review · Reviewer_NWLb · 2025-03-10

**Overall Recommendation:** 4

**Summary:**

This paper presents the FlexiReID framework, which addresses the issue of modality combination retrieval that has been largely overlooked in the current cross-modal person re-identification (ReID) field. Specifically, traditional approaches in cross-modal ReID typically use a single modality as the query to match the corresponding person. This paper introduces, for the first time, the concept of flexible combination retrieval, where the query can be replaced with different modality combinations (e.g., text, sketch, infrared), supporting seven distinct retrieval methods, thereby enhancing the model's applicability across diverse scenarios.
The approach proposed in this paper includes the innovative introduction of an adaptive mixture of experts (MoE) mechanism, applied during the modality feature extraction process. In contrast to the traditional Top-K routing mechanism used in MoE models, the adaptive routing mechanism introduced here allows for flexible selection of the number of experts based on the complexity of the features. To ensure that the model selects only the minimal number of necessary experts, the paper also incorporates an adaptive loss function. This enables the model to handle different modality extraction tasks without increasing computational burden.
Additionally, this paper introduces a novel feature fusion method, CMQF, which employs a two-layer fusion structure consisting of an interaction layer and a fusion layer. To address the issue of missing modalities, the method uses learnable feature vectors as substitutes, thereby effectively considering all possible modality combinations. The final model is optimized using contrastive loss.
In the experimental validation section, the paper constructs the CIRS-PEDES dataset, spanning four modalities. Compared to methods that only support single modality cross-modal retrieval, FlexiReID achieves optimal performance. Furthermore, the combination retrieval methods supported by FlexiReID further enhance retrieval performance, demonstrating that the use of multiple modalities for person re-identification leads to superior results.

**Claims And Evidence:**

I believe the effectiveness of the FlexiReID method proposed in the paper is indeed validated through experiments on multiple datasets. Specifically, tests were conducted on four datasets with expanded modalities, evaluating seven different person re-identification methods. The results show that, in single-modality retrieval tasks, FlexiReID achieves optimal performance compared to the current state-of-the-art cross-modal ReID methods. Moreover, FlexiReID supports combination modality retrieval methods, which are not supported by mainstream approaches. It also demonstrates that using multiple modality combinations as queries can lead to superior performance, as opposed to relying on a single modality as the query. This highlights the significant practical implications of FlexiReID, which enables flexible matching of persons across various modalities in real-world applications.

**Essential References Not Discussed:**

I believe the FlexiReID method proposed in this paper unifies various traditional cross-modal ReID tasks within a single model, while also supporting flexible combination modality retrieval. This significantly enhances the versatility of a single model. The concept of flexible combination modality retrieval has not been addressed in previous ReID literature, and this method opens up a new exploratory direction in the field of ReID.

**Experimental Designs Or Analyses:**

I have thoroughly reviewed the experimental design and analysis section of this paper, and I believe it is generally well-reasoned. In the performance evaluation section, the paper presents results from models evaluated on four datasets spanning RGB, sketch, infrared, and text modalities, and compares them with current state-of-the-art cross-modal ReID methods. Overall, FlexiReID achieves optimal performance across various cross-modal ReID tasks on different datasets. Additionally, the combination modality retrieval methods it supports further enhance its performance.
In terms of ablation experiments, the paper conducts ablations on the adaptive MOE and feature fusion modules, as well as on the two hyperparameters: the number of experts and the threshold confidence. I believe the ablation experiments could be further supplemented by including an ablation of the adaptive routing mechanism versus the Top-K mechanism, which would provide a more comprehensive demonstration of the advantages of the proposed adaptive MOE.

**Methods And Evaluation Criteria:**

I believe the FlexiReID framework proposed in this paper holds significant importance for the development of the person re-identification (ReID) field. ReID has evolved from single-modality matching to cross-modal matching, during which many excellent recognition methods have emerged. However, these methods all belong to the one-to-one identification category, where a model supports only one modality as a query to match another modality. This paper transcends the traditional one-to-one ReID paradigm by introducing the concept of many-to-one, where multiple modality combinations serve as queries to match the target modality, thus opening up a new research direction. Moreover, with the advancement of various surveillance technologies, acquiring multiple modalities of pedestrians has become increasingly common. Therefore, the need for a generalizable model that supports both cross-modal and combination modality retrieval has become more pressing. For these reasons, I consider the FlexiReID method proposed in this paper to be of great significance for complex real-world applications. On the other hand, this paper also expands current ReID datasets by incorporating additional modalities. I believe this creates the necessary data conditions for the emerging research direction of combination retrieval.

**Other Comments Or Suggestions:**

I believe that the greatest contribution of this article is the proposal of the concept of combinatorial modal person re-identification, which caters to the era of the increasing development of various monitoring technologies. The constructed FlexiReID framework, as a unified model integrating various cross-modal ReID tasks, has strong practical significance. In addition, the proposed adaptive MOE method is also quite innovative.
It is hoped that in the future, the performance of the adaptive routing mechanism can be compared with that of the Top - K routing mechanism to further demonstrate its advantages.

**Other Strengths And Weaknesses:**

Strengths：
1）This paper introduces the concept of combination modality person re-identification (ReID) for the first time, opening up a new research direction in the ReID field. I believe that this approach, which integrates various cross-modal ReID tasks into a single framework, holds significant importance for complex real-world application scenarios.
2）This paper proposes the adaptive MOE method, which, in contrast to traditional MOE methods based on the Top-K routing mechanism, offers a more flexible expert selection strategy. By introducing adaptive loss, this module can automatically learn and select the minimum number of necessary experts, further optimizing both the model's performance and computational cost.
3）I appreciate the illustrations in this paper; they are clear and highly readable. Each figure effectively aids in understanding the method proposed by the authors, allowing readers to quickly grasp the key concepts. The details and annotations in the figures are also well-executed, enhancing both the professionalism and comprehensibility of the illustrations. Overall, the design of the figures significantly contributes to the presentation of the paper, greatly improving its overall readability.

Weaknesses
1）The ablation experiment section lacks a comparison between the adaptive routing mechanism and the Top-K routing mechanism. It would be helpful to include this comparison.
2）The paper could explore different feature fusion methods, test their performance, and compare them with the CMQF feature fusion method proposed in the paper to highlight the advantages of the proposed approach.

**Questions For Authors:**

1）The paper proposed a two - layer feature fusion method and achieved good results. Could you try to compare other feature fusion methods and illustrate the advantages of the method you proposed?
2）Could you conduct an additional ablation experiment to compare the performance of the adaptive routing mechanism you proposed with that of the traditional Top - K routing mechanism?

**Relation To Broader Scientific Literature:**

The concept of flexible combination modality retrieval proposed in this paper represents a further development of previous cross-modal person re-identification (ReID) work. Typical examples of traditional cross-modal ReID tasks include text-to-visible modality and infrared-to-visible modality person re-identification. A classic paper on text-to-visible person re-identification is Cross-Modal Implicit Relation Reasoning and Aligning for Text-to-Image Person Retrieval, which introduces an implicit reasoning and alignment model called IRRA. This model leverages cross-modal implicit local relation learning for global alignment without requiring any additional supervision or reasoning costs.
On the other hand, for infrared-to-visible re-identification, the paper Multi-Stage Auxiliary Learning for Visible-Infrared Person Re-Identification proposes MSALNet, which first applies grayscale histogram transformations to infrared and visible light images and trains the model in two stages to reduce color-related effects. It then uses the HFCL module to fuse cross-modal information and the MSR module to suppress low-similarity feature locations. Finally, the DCA loss function is used to optimize the distance between samples and cross-modal class centers, reducing intra-class variation.
I believe the framework proposed in this paper not only encompasses the functionality of traditional ReID tasks but also incorporates the capability for combination modality retrieval, which previous frameworks lacked, thus enhancing its versatility.

**Theoretical Claims:**

The method proposed in this paper is generally sound. Specifically, the paper applies two innovative techniques: adaptive mixture of experts (MOE) and modality fusion. Regarding adaptive MOE, the paper proposes an adaptive routing mechanism to overcome the limitation of the traditional MOE model, which uses the Top-K routing mechanism to select a fixed number of experts. The proposed adaptive routing mechanism customizes the number of experts based on the complexity of the input modality features.
As for the modality fusion technique, it employs a two-layer structure, consisting of an interaction layer and a fusion layer, with ablation experiments providing strong evidence of its effectiveness. I believe the authors could further explore additional modality fusion strategies to better highlight the advantages of their proposed method.

---

> ### Author Rebuttal · Authors · 2025-03-31
>
> > The ablation experiment section lacks a comparison between the adaptive routing mechanism and the Top-K routing mechanism. It would be helpful to include this comparison.
> >
>
> A1: In fact, we have already compared a method using the Top-K mechanism in our ablation study, specifically in Row No.1 of Table 3. Additionally, we have supplemented our work with comparative experiments of CMQF combined with other routing mechanisms(see Table 3 in our response to reviewer YSVq). These ablation studies on routing strategies further validate the effectiveness of our proposed adaptive routing.
>
> > The paper could explore different feature fusion methods, test their performance, and compare them with the CMQF feature fusion method proposed in the paper to highlight the advantages of the proposed approach.
> >
>
> A2: Following your suggestion, we introduced three fusion strategies for comparison with our CMQF, as shown in the table below. The three strategies are: Concatenation—concatenating features from different modalities and feeding them into a Transformer module for fusion; Summation—summing features from different modalities and then using a Transformer for fusion; and Hierarchical Fusion—passing each modality through its own Transformer module first, followed by a shared Transformer for final fusion. It can be seen from Table 1 below that our CMQF achieves the best performance among all methods.
>
> Table1: Comparative experiment of feature fusion methods
>
> | No | Method | T—R | S—R | IR—R | T+S—R | T+IR—R | S+IR—R | T+S+IR—R | Avg. |
> | --- | --- | --- | --- | --- | --- | --- | --- | --- | --- |
> | 1 | Concatenation | 68.87 | 84.13 | 84.37 | 86.41 | 85.93 | 84.97 | 87.35 | 83.14 |
> | 2 | Summation | 67.85 | 83.20 | 83.51 | 85.42 | 85.06 | 84.08 | 86.54 | 82.24 |
> | 3 | Hierarchical Fusion | 68.93 | 84.47 | 84.82 | 86.94 | 86.11 | 85.55 | 87.76 | 83.51 |
> | 4 | CMQF(Ours) | 69.20 | 84.92 | 85.26 | 87.47 | 86.23 | 85.97 | 88.23 | 83.90 |

---

### Official Review · Reviewer_Tfk2 · 2025-03-13

**Overall Recommendation:** 3

**Summary:**

This paper firstly introduces  the concept of flexible retrieval in the field of person re-identification and propose a corresponding method FlexiReID which supports flexible retrieval with arbitrary modality combinations. The authors also constructed a unified dataset by existing ReID datasets.

**Claims And Evidence:**

The concept is reasonable.

**Essential References Not Discussed:**

NA

**Experimental Designs Or Analyses:**

I am curious about the comparative experiment. This paper only provides the comparison with SOTA methods on Text-to-RGB task. However,   there exists previous methods for other dual modalies retrieval. But this paper do not provide the comparison results.

**Methods And Evaluation Criteria:**

The authors propose AEA-MoE mechanism to dynamically selects different numbers of experts and CMQF module to leverage learnable embedding features to compensate for missing modalities and fuse different modality features. It seems reasonable to solve this new problem.
This paper provide the widely used metrices like Rank-K accuracy、mAP、mINP.

**Other Comments Or Suggestions:**

NA

**Other Strengths And Weaknesses:**

Strengths:
The method is practical and has clear application value.

Weakness:
The author used generative models (StyleGAN3, InfraGAN, GPT-4) to extend the modality of sketches, infrared images, and text descriptions. Although this approach is effective, it may result in significant differences between the generated data and the actual collected data.

**Questions For Authors:**

NA

**Relation To Broader Scientific Literature:**

This work is releated to  Cross-modal Person Re-identification, Mixture-of-Experts and Vision-Language Pre-training Models.

**Theoretical Claims:**

This article does not involve strict theoretical proofs, nor does it propose clear theoretical propositions or theorems. The main contributions in the article are focused on method design, module implementation, experimental verification, and qualitative analysis, so there is no issue of reviewing theoretical proofs.

---

> ### Author Rebuttal · Authors · 2025-03-31
>
> > I am curious about the comparative experiment. This paper only provides the comparison with SOTA methods on Text-to-RGB task. However, there exists previous methods for other dual modalies retrieval. But this paper do not provide the comparison results.
> >
>
> A1: Thank you for the valuable question. Currently, there are still few works focusing on multi-modal retrieval. Among them, UNI-ReID is one of the representative methods that support retrieval across multiple modalities (text, sketch, and their combination). Our work is the first to systematically explore the novel query paradigm of *flexible compositional retrieval*, and we are the first to propose the many-to-one retrieval setting. Therefore, there are no existing methods that can be directly compared with ours under this setting. If the reviewer has any recommended works (preferably with open-source code，due to time constraints), we would be glad to include them in our discussion.
>
> Nonetheless, we have conducted comprehensive comparisons with existing methods on several dual-modality retrieval tasks. For example, we compared with recent Sketch-to-RGB methods on the PKU-Sketch dataset (see Table 1 in our response to reviewer dFYK), and with recent Infrared-to-RGB methods on the RegDB dataset (see Table 4 in our response to reviewer YSVq). In addition, we have included the latest methods for the S→R and IR→R tasks in Table 1 of the paper (see Table 1 in our response to reviewer YSVq). These results demonstrate that FlexiReID achieves competitive performance across multiple tasks, further validating its advantages as a unified and flexible framework for multimodal retrieval.
>
> > The author used generative models (StyleGAN3, InfraGAN, GPT-4) to extend the modality of sketches, infrared images, and text descriptions. Although this approach is effective, it may result in significant differences between the generated data and the actual collected data.
> >
>
> A2: Thank you for the valuable question. It is true that using generative models such as StyleGAN3, InfraGAN, and GPT-4 to synthesize modalities like sketches, infrared images, and textual descriptions may introduce certain differences compared to real-world collected data. However, this approach is particularly meaningful at the current stage, as publicly available person re-identification datasets spanning three or more modalities are extremely limited. The inclusion of synthetic modalities greatly enriches the diversity of data modality, facilitates the construction of a unified multimodal dataset, and provides essential support for training our proposed flexible compositional retrieval framework.
>
> To assess the practical effectiveness of our method, we have also conducted evaluations on several real-modality datasets. For instance, as shown in Table 2 of the article, SYSU-MM01 is a real NIR-to-RGB retrieval dataset. Additionally, we evaluated our method on the PKU-Sketch dataset, which contains real sketch modality data (see Table 1 in our response to reviewer dFYK), and on RegDB, a dataset with real infrared images (see Table 4 in our response to reviewer YSVq). These experiments further validate the effectiveness and applicability of FlexiReID in real-world scenarios.
>
> We believe FlexiReID serves as a solid and flexible foundation, which can be further fine-tuned using real-world data to enhance its adaptability. While some discrepancies exist between synthetic and real data, this reflects a necessary stage in the research process. Our work offers new insights and methodologies for multimodal person re-identification and lays a foundation for future practical deployment.

---

### Official Review · Reviewer_YSVq · 2025-03-13

**Overall Recommendation:** 3

**Summary:**

The paper proposes the FlexiReID framework to support person retrieval across seven different modality combinations (such as text, sketches, infrared images, RGB images, and their combinations). The framework comprises an AEA-MoE mechanism for dynamically selecting varying numbers of expert networks according to input features, and a CMQF module that is capable of effectively integrating features from different modalities and compensating for missing modalities through learnable embedding features. To support the study, the paper constructs a dataset named CIRS-PEDES which unifies four modalities. Extensive experiments show FlexiReID's efficacy in multimodal person re-identification.

**Claims And Evidence:**

No.
In L68-71, the claim  ’ which supports flexible retrieval with arbitrary modality combinations’ is problematic. FlexiReID only supports seven different modality combinations and misses the other modalities such as thermal, LiDAR and event data.

**Essential References Not Discussed:**

No.

**Experimental Designs Or Analyses:**

In Table 1, except for T-R task, there are few comparison results for other task, which cannot demonstrate the superiority of the proposed method. Similarly, in Table 2,  there are no comparison results for six tasks. And the compared methods for IR-R task are not SOTA, the proposed method demonstrates relatively average performance on the dataset.

**Methods And Evaluation Criteria:**

No.
The paper misses model comparison experiments for multi-modal retrieval on  benchmark datasets.

**Other Comments Or Suggestions:**

It is suggested to expand Table 1 and Table 2 by adding additional comparative results, particularly from more recent studies. As for ablation study, it is suggested to compare CMQF with other routing mechanism.

-----------------
The author addressed most of my comments, so I will raise my score.

**Other Strengths And Weaknesses:**

Strength:
The paper introduces, for the first time, the concept of flexible retrieval, which supports seven different modality combinations for retrieval.

Weakness:
Based on the experimental results presented in the paper, the performance of the proposed method is relatively modest.

**Questions For Authors:**

1.	The paper focuses on unified cross-modal person re-identification framework, could you explain the motivation behind this idea? Given that RGB -RGB is the most common task, why not integrate it into a more unified framework?
2.	The paper employs generative models to synthesize missing modalities, could you assess the impact of synthetic data on model performance through comparative experiments? And it would be more convincing to evaluate the approach on other real-life datasets.
3.	Could you also analyze the computational complexity and inference speed of the proposed method?

**Relation To Broader Scientific Literature:**

The key contributions of the paper related to unified person ReID model that can handle multi retrieval task in a unified modal, such as UNIReID

**Theoretical Claims:**

No proofs.

---

> ### Author Rebuttal · Authors · 2025-03-31
>
> Thanks for your thoughtful feedback. We'll address each of your concerns in detail.
>
> > No. The paper misses model comparison experiments for multi-modal retrieval on benchmark datasets.
> >
>
> > In Table 1, except for T-R task, there are few comparison results for other task, which cannot demonstrate the superiority of the proposed method. Similarly, in Table 2, there are no comparison results for six tasks. And the compared methods for IR-R task are not SOTA, the proposed method demonstrates relatively average performance on the dataset.
> >
>
> > Based on the experimental results presented in the paper, the performance of the proposed method is relatively modest.
> >
>
> A1: There are few models for multi-modal retrieval, with UNIReID being one of the representative works. Our work is the first to explore flexible compositional retrieval, so no existing methods are directly comparable for the many-to-one setting. If you have any recommended works (preferably with open-source code，due to time constraints), we’d be glad to include them. Nevertheless, We provide additional comparisons with S-R and IR-R methods in Table 1 below, and report UNIReID’s performance on three tasks in Table 2 below. It is important to note that we did not employ any task-specific module designs or training strategies for individual retrieval tasks, which may limit performance in single-task scenarios. However, compared to traditional single-modality cross-modal retrieval, FlexiReID supports a wider range of retrieval modes and demonstrates better generalization capabilities.
>
> Table1: Supplementary experiments on CUHK-PEDES
>
> | Tasks | Methods | Venue | R1 | mAP |
> | --- | --- | --- | --- | --- |
> | S—R | Sketch Trans+ | PAMI2023 | 81.39 | 73.72 |
> |  | DALNet | AAAI2024 | 83.03 | 75.39 |
> |  | FlexiReID(Ours) | - | 84.92 | 79.21 |
> | IR—R | GUR | ICCV2023 | 82.06 | 75.84 |
> |  | SDCL | CVPR2024 | 84.57 | 77.32 |
> |  | FlexiReID(Ours) | - | 85.26 | 79.43 |
>
> Table2: Supplementary experiments on SYSU-MM01
>
> |  | SYSU-MM01 | ALL-Search |  | Indoor-Search |  |
> | --- | --- | --- | --- | --- | --- |
> | Tasks | Methods | R1 | mAP | R1 | mAP |
> | T—R | UNIReID | 54.6 | 52.8 | 56.3 | 63.5 |
> |  | FlexiReID(Ours) | 56.8 | 65.4 | 58.2 | 67.6 |
> | S—R | UNIReID | 64.2 | 57.7 | 65.8 | 73.8 |
> |  | FlexiReID(Ours) | 66.4 | 60.3 | 68.5 | 75.3 |
> | T+S—R | UNIReID | 66.9 | 65.9 | 67.9 | 72.7 |
> |  | FlexiReID(Ours) | 68.7 | 67.2 | 70.6 | 73.4 |
>
> > It is suggested to compare CMQF with other routing mechanism.
> >
>
> A2: We added comparisons between CMQF and other routing mechanisms (Table 3). Our adaptive routing shows better average performance than the alternatives.
>
> Table3: Comparison experiment of routing mechanisms
>
> | Routing | Avg. |
> | --- | --- |
> | Top-K | 80.58 |
> | Soft Routing | 81.80 |
> | Hash Routing | 83.11 |
> | Ours(Adaptive Routing) | 83.90 |
>
> > The paper focuses on unified cross-modal person re-identification framework, could you explain the motivation behind this idea? Given that RGB -RGB is the most common task, why not integrate it into a more unified framework?
> >
>
> A3: Our unified framework aims to handle diverse real-world inputs beyond fixed modality pairs (e.g., Text-RGB, IR-RGB). In practice, users may provide multiple modalities (e.g., text, sketch, IR), which existing models struggle to integrate. FlexiReID supports seven modality combinations, improving retrieval performance and robustness. Following your suggestion, we also added RGB-to-RGB retrieval, with results on Market-1501 and MSMT17 shown in Table 2 of our response to reviewer dFYK.
>
> > Could you assess the impact of synthetic data on model performance through comparative experiments? And it would be more convincing to evaluate the approach on other real-life datasets.
> >
>
> A4: We use generative models to fill missing modalities, enabling flexible retrieval despite incomplete dataset coverage. To evaluate the impact of synthetic data, we tested on real-world datasets including SYSU-MM01 (Table 2 in the article), RegDB (Table 4 below), and PKU-Sketch (Table 1 in our response to reviewer dFYK). The results confirm FlexiReID’s practical effectiveness as a unified, adaptable framework with strong real-world deployment potential.
>
> Talbe4: Experiments on RegDB(IR-R)
>
> | Method | Venue | R1 | mAP |
> | --- | --- | --- | --- |
> | SFANet | TNNLS23 | 70.2 | 63.8 |
> | CAJ | TPAMI24 | 84.9 | 77.8 |
> | DARD | TIFS24 | 85.5 | 85.1 |
> | FlexiReID | - | 88.6 | 87.4 |
>
> > Could you also analyze the computational complexity and inference speed of the proposed method?
> >
>
> A5: We analyzed the computational complexity and inference speed of FlexiReID. With a frozen CLIP encoder and lightweight AEA-MoE and CMQF modules, it runs at 19 GFLOPs and 14 ms/query on a single NVIDIA 3090 GPU—comparable to UNIReID (17 GFLOPs, 11 ms/query), which supports only three fixed modality combinations. In contrast, FlexiReID supports seven flexible combinations with similar efficiency, making it more adaptable and deployable.

---

### Official Review · Reviewer_dFYK · 2025-03-17

**Overall Recommendation:** 3

**Summary:**

FlexiReID is a novel framework for multimodal person re-identification that enables flexible retrieval across various single or combined modalities—including text, sketches, RGB, and infrared images—thereby addressing the limitations of existing methods that focus on only one or two modality pairs. By introducing an adaptive mixture of experts (MOE) mechanism, FlexiReID dynamically integrates outputs from different expert networks, leveraging each modality’s strengths to enhance retrieval performance. Additionally, a cross-modal query fusion module refines the fused features to optimize their representational quality. To evaluate the framework comprehensively, the authors construct a unified dataset called CIRS-PEDES, derived from four existing ReID datasets (CUHK-PEDES, ICFG-PEDES, RSTPReID, and SYSU-MM01) and enriched with text, sketches, RGB, and infrared data.

**Claims And Evidence:**

Yes

**Essential References Not Discussed:**

None

**Experimental Designs Or Analyses:**

Table1 illustrates FlexiReID achieves promising accuracy in the T+S+IR→R situations. However these sketches are generated from RGB images, while in real-world scenarios, sketches is a front-facing portrait sketched from memory, which leads to a substantial discrepancy between the sketches depicted in the paper and the official version.

**Methods And Evaluation Criteria:**

Yes

**Other Comments Or Suggestions:**

None

**Other Strengths And Weaknesses:**

Strengths:
1. The paper considers muliti-modal person re-ID and the proposed AEA-MOE is proved to be effective in muliti-modal person re-ID.

Weaknesses:
1. In Line433, the authors claims "No.2 employs the traditional MOE method, while No.3 utilizes AEA-MOE." which seems not consistent with Table 3?
2. The paper focuses on the multi-modal person re-ID, but these multi-modalities are generated via AI tools which is not align with real-world scenarios. Thus, there remains a gap compared to practical applications.

**Questions For Authors:**

1. The authors propose a flexible multimodal re-ID framework, but why do the authors ignore RGB-to-RGB retrieve, which is the main stream of re-ID?
2. In Figure2, why do the authors apply SDM loss to features of the same images, in which the model cannot retrieve persons across various situations?

**Relation To Broader Scientific Literature:**

This work is related to multimodal retrieval, demonstrating that combining multiple modalities can lead to stronger feature embeddings and higher accuracy.

**Theoretical Claims:**

There do not exist theoretical claims.

---

> ### Author Rebuttal · Authors · 2025-03-31
>
> Thanks for your careful and valuable comments. We will explain your concerns point by point.
>
> > Table1 illustrates FlexiReID achieves promising accuracy in the T+S+IR→R situations. However these sketches are generated from RGB images, while in real-world scenarios, sketches is a front-facing portrait sketched from memory, which leads to a substantial discrepancy between the sketches depicted in the paper and the official version.
> >
>
> > The paper focuses on the multi-modal person re-ID, but these multi-modalities are generated via AI tools which is not align with real-world scenarios. Thus, there remains a gap compared to practical applications.
> >
>
> A1: In response to the two similar concerns you raised, we provide the following explanation. It is indeed true that sketches and other multimodal data generated using intelligent tools may differ from those in real-world scenarios. However, this approach still holds significant value. Currently, publicly available person re-identification datasets that span more than two modalities are scarce. Leveraging such generative methods can substantially enrich the modality diversity of datasets, facilitating the construction of unified multimodal datasets and providing more comprehensive support for model training.Moreover, we have also conducted experiments on real-world test sets. For instance, as shown in Table 2, SYSU-MM01 is a real near-infrared (NIR) to RGB retrieval dataset. The inclusion of generated modalities led to notable performance improvements. In addition, we evaluated our method on the PKU-Sketch dataset, which contains real sketch modality data (as shown in the table1 below), and on the RegDB dataset, which includes real infrared modality data (see Table 4 in our response to reviewer YSVq), further validating the effectiveness of our approach in real-world applications.
>
> We believe that FlexiReID can serve as a foundational framework, which can be further fine-tuned with data from real-world scenarios to enhance the model's adaptability to practical environments. Therefore, although some modality discrepancies exist at this stage, this reflects a common characteristic of research in its developmental phase. Our work introduces new perspectives and methodologies for the field of multimodal person re-identification and lays a solid foundation for future research in real-world applications.
>
> Table 1: Experiments on PKU-Sketch
>
> | Methods | Reference | mAP | Rank@1 | Rank@5 | Rank@10 |
> | --- | --- | --- | --- | --- | --- |
> | CCSC | MM22 | 83.7 | 86.0 | 98.0 | 100.0 |
> | Sketch Trans+ | PAMI2023 | - | 85.8 | 96.0 | 99.0 |
> | DALNet | AAAI2024 | 86.2 | 90.0 | 98.6 | 100.0 |
> | FlexiReID（OUrs） | - | 91.2 | 93.5 | 99.3 | 100.0 |
>
> > In Line433, the authors claims "No.2 employs the traditional MOE method, while No.3 utilizes AEA-MOE." which seems not consistent with Table 3?
> >
>
> A2: Thank you for your correction. There was indeed a labeling error in the manuscript. In fact, No.0 corresponds to the zero-shot CLIP backbone baseline, No.1 adopts the conventional MoE approach based on the Top-K mechanism, while No.2 and No.3 employ the proposed AEA-MoE method. We will make the necessary corrections in the revised version.
>
> > The authors propose a flexible multimodal re-ID framework, but why do the authors ignore RGB-to-RGB retrieve, which is the main stream of re-ID?
> >
>
> A3: Thank you for your constructive suggestion. Following your advice, we have included RGB-to-RGB retrieval in our evaluation. We assessed the performance of FlexiReID on the MSMT17 and Market-1501 dataset, and the corresponding results are presented in the table2 below:
>
> Table 2: Experiments on Market-1501 and MSMT17
>
> |  | Market-1501 |  | MSMT17 |  |
> | --- | --- | --- | --- | --- |
> | Methods | Rank@1 | mAP | Rank@1 | mAP |
> | FastReID(ACMMM23) | 95.4 | 88.2 | 83.3 | 59.9 |
> | BPBreID(WACV23) | 95.1 | 87.0 | - | - |
> | MVI2P(Inf Fusion24) | 95.2 | 87.0 | 80.4 | 56.4 |
> | FlexiReID(Ours) | 96.0 | 92.1 | 83.7 | 67.5 |
>
> > In Figure2, why do the authors apply SDM loss to features of the same images, in which the model cannot retrieve persons across various situations?
> >
>
> A4: I may have misunderstood your question. Are you asking why the SDM loss is applied to images of pedestrians with the same pose in the same scene during training? In fact, during the data processing stage of training, we construct modality pairs using images of the same identity captured in different scenes and with different poses, then compute the SDM loss based on these pairs. During testing, the model is also capable of retrieving images of the same pedestrian taken in different scenes from the query. We will refine the illustration in Figure 2 to eliminate this ambiguity. If you have any further questions, we look forward to continued discussions.

---

### Decision · Program_Chairs · 2025-05-01

**Decision:**

Accept (poster)

**Comment:**

Five experts in the field reviewed this paper. Their recommendations are 3 Weak Accepts and 2  Accepts. Overall, the reviewers appreciated the paper because the new FlexiReID method integrates an adaptive MOE and CMQF feature modality fusion. The AEA-MOE mechanism dynamically selects different numbers of experts. It can be versatile and effective in multimodal person re-ID, and supports seven different modality combinations for retrieval. Moreover, the results are convincing, showing that FlexiReID can achieve promising accuracy.
   Some reviewers initially raised concerns regarding the paper, but the author provided a detailed and insightful response that addressed most of these concerns. For example, there is concern that sketches and other multimodal data generated using AI tools (StyleGAN3, InfraGAN, GPT-4) to synthesize missing modalities that may not align with those captured in real-world scenarios. However, the authors argue that multimodal datasets are limited, and have shown that such generative methods can improve the diversity of datasets, and model accuracy.
    I recommend accepting this paper based on the reviewers’ feedback and the authors’ satisfactory rebuttal. Overall, FlexiReID is motivated, holds significant novelty, and the experimental validation is convincing. However, the reviewers have raised some concerns that should be addressed in the final camera-ready version of the paper. In particular, the ablations lacks comparisons with different adaptive routing mechanisms and feature fusion methods. This paper only provides the comparison with SOTA methods on cross-modal ReID tasks. Based on the experimental results, the performance of the proposed method may be relatively modest.